# Rough and smooth variants of *Mycobacterium abscessus* are differentially controlled by host immunity during chronic infection of adult zebrafish

Julia Y. Kam[1], Elinor Hortle[1,2], Elizabeth Krogman[1], Sherridan E. Warner[1,2], Kathryn Wright [1], Kaiming Luo[1], Tina Cheng [1], Pradeep Manuneedhi Cholan[1], Kazu Kikuchi[3,4], James A. Triccas [2], Warwick J. Britton [1,5], Matt D. Johansen [6], Laurent Kremer [6,7] & Stefan H. Oehlers [1,2,8 ✉]

Prevalence of *Mycobacterium abscessus* infections is increasing in patients with respiratory comorbidities. After initial colonisation, *M. abscessus* smooth colony (S) variants can undergo an irreversible genetic switch into highly inflammatory, rough colony (R) variants, often associated with a decline in pulmonary function. Here, we use an adult zebrafish model of chronic infection with R and S variants to study *M. abscessus* pathogenesis in the context of fully functioning host immunity. We show that infection with an R variant causes an inflammatory immune response that drives necrotic granuloma formation through host TNF signalling, mediated by the *tnfa*, *tnfr1* and *tnfr2* gene products. T cell-dependent immunity is stronger against the R variant early in infection, and regulatory T cells associate with R variant granulomas and limit bacterial growth. In comparison, an S variant proliferates to high burdens but appears to be controlled by TNF-dependent innate immunity early during infection, resulting in delayed granuloma formation. Thus, our work demonstrates the applicability of adult zebrafish to model persistent *M. abscessus* infection, and illustrates differences in the immunopathogenesis induced by R and S variants during granulomatous infection.

[1] Tuberculosis Research Program at the Centenary Institute, The University of Sydney, Camperdown, NSW, Australia. [2] The University of Sydney, Faculty of Medicine and Health & Marie Bashir Institute, Camperdown, NSW, Australia. [3] Developmental and Stem Cell Biology Division, Victor Chang Cardiac Research Institute, Darlinghurst, NSW, Australia. [4] St. Vincent's Clinical School, University of New South Wales, Kensington, NSW, Australia. [5] Department of Clinical Immunology, Royal Prince Alfred Hospital, Camperdown, NSW, Australia. [6] Centre National de la Recherche Scientifique UMR 9004, Institut de Recherche en Infectiologie de Montpellier (IRIM), Université de Montpellier, Montpellier, France. [7] INSERM, IRIM, Montpellier, France. [8] A*STAR Infectious Diseases Labs, Agency for Science, Technology and Research (A*STAR), Singapore, Singapore. ✉email: stefan_oehlers@idlabs.a-star.edu.sg

Mycobacterium abscessus is an increasingly recognized human pathogen responsible for a wide array of clinical manifestations including muco-cutaneous infections and disseminated or chronic pulmonary diseases[1]. The latter is mostly encountered in patients with underlying lung disorders, such as bronchiectasis or cystic fibrosis (CF). Irrespective of being a rapid-growing mycobacteria (RGM), M. abscessus displays many pathophysiological traits with slow-growing mycobacteria (SGM), such as Mycobacterium tuberculosis. These include the capacity to persist silently within granulomatous structures and to produce pulmonary caseous lesions[2,3]. In addition, M. abscessus is notorious for being one of the most-drug resistant mycobacterial species, characterized by a wide panel of acquired and innate drug resistance mechanisms against nearly all anti-tubercular drugs, as well as many different classes of antibiotics[1,4]. Consequently, this explains the complexity and duration of the treatments and the high level of therapeutic failure[5].

M. abscessus exists either as smooth (S) or a rough (R) colony morphotype variants associated with distinct clinical outcomes[6]. Previous epidemiological studies have highlighted the association of the R variant that can persist for many years in the infected host, with a rapid decline in pulmonary function[7–9]. It is well established that these morphological differences between S and R variants are dependent on the presence or absence of surface-exposed glycopeptidolipids (GPL), respectively, which are reduced or lost in R variants by transcriptional downregulation or loss of function mutations[6,10–14]. However, our knowledge of the pathophysiological characteristics and interactions between R or S variants with the host immune cells remains largely incomplete and is hampered by the lack of animal models that are permissive to persistent M. abscessus infection[15].

Intravenous injection or aerosol administration of M. abscessus in immunocompetent BALB/c mice fails to establish a persistent infection, typified by the rapid clearance of the bacilli from the liver, spleen and lungs within 4 weeks[16]. Immunosuppression is required to produce a progressive high level of infection with M. abscessus in mice, either genetically, as evident in nude, severe combined immunodeficiency (SCID), interferon-gamma (GKO) and granulocyte-macrophage colony-stimulating factor (GM-CSF) knock-out mice, or chemically by dexamethasone treatment[17,18]. Recently, an improved model of intratracheal agar bead inoculation has been adapted for M. abscessus causing a persistent infection for at least two months post implantation[17,19].

The contribution of B and T cells in the control of M. abscessus infection has been studied in C57BL/6 mice with Rag2$^{-/-}$, Cd3e$^{-/-}$ and μMT$^{-/-}$ knockouts[20]. These studies indicated that infection control was primarily T cell-dependent in the spleen, and both B and T cell-dependent in the liver. In addition, IFNγ-receptor KO mice (ifngr1$^{-/-}$) were found to be significantly impaired in their control of M. abscessus both in the spleen and in the liver, as were TNF$^{-/-}$ mice that had more pronounced and markedly dysregulated granulomas[20]. Collectively, these findings highlight the central role of T cell immunity, IFNγ and TNF for the control of M. abscessus in C57BL/6 mice, similarly to the control of M. tuberculosis infection.

In recent years, alternative non-mammalian models, such as Drosophila[21], Galleria larvae[22], and zebrafish embryos[15] have been developed to study the chronology and pathology of M. abscessus infection and for in vivo therapeutic assessment of drugs active against M. abscessus. In particular, zebrafish embryos have delivered important insights into the pathogenesis of M. abscessus and the participation of innate immunity in controlling infection[10,23]. The optical transparency of zebrafish embryos has been used to visualise the formation of large extracellular cords by the R form in vivo, representing a mechanism of immune subversion by preventing phagocytic destruction and highlighting the

importance of mycobacterial virulence factors such as the dehydratase MAB_4780 and the MmpL8$_{MAB}$ lipid transporter[10,24,25]. Other studies in zebrafish embryos have demonstrated the contribution of host TNF signalling and IL8-mediated neutrophil recruitment for protective granulomatous immunity against M. abscessus[23], and the link between dysfunctional cystic fibrosis transmembrane conductance regulator (CFTR) and vulnerability to M. abscessus infection via phagocytic oxidative responses[26].

Adult zebrafish models have been well-described for the study of mycobacterial pathogenesis by Mycobacterium marinum, used as a surrogate for the closely related M. tuberculosis, and the human pathogen Mycobacterium leprae[27–30]. Encompassing a fully functional immune system, previous studies in adult zebrafish with pathogenic mycobacteria, such as M. marinum, have unravelled the interplay between innate and adaptive immunity in mycobacterial granuloma formation and function.

Here, we show adult zebrafish are a useful host to analyse and compare the chronology of infection with M. abscessus S and R variants and to study the contribution of the T cell-mediated immunity in the granulomatous response in M. abscessus infection.

## Results

**Adult zebrafish can be chronically infected with *M. abscessus*.** We infected adult zebrafish with approximately $10^5$ CFU per animal with the rough (R) and smooth (S) variants of the reference strain CIP104536$^T$, scaled for the smaller size of zebrafish from $10^6$–$10^7$ used in mouse intravenous infections[16,17,20]. To determine if M. abscessus produces a persistent infection in adult zebrafish, we performed CFU recovery on animals across 28 days of infection (Fig. 1A). Variation in the initial inoculum ranging from $10^4$ to $10^6$ CFU did not appear to impact the course of infection burden with stable burden of the R variant within a 1-log window either side of the inoculation dose up to 28 days post infection (dpi) and progressive growth of the S variant to approximately 1-log above the inoculation dose at 28 dpi in each of three experiments.

Normalising burdens across three independent experiments per M. abscessus variant to perform statistical testing, we observed statistically significant increases in the proliferation of M. abscessus S compared to R variant at 7, 14, and 28 dpi (Fig. 1B). Furthermore, comparison of the Day 0 and Day 28 burdens demonstrated M. abscessus R burdens were statistically unchanged ($P > 0.99$, ANOVA) while M. abscessus S burdens increased by approximately 20-fold across the 4 weeks ($P = 0.024$, ANOVA).

**Granuloma histopathology is accelerated during *M. abscessus* R infection compared to S.** We next performed histology on adult zebrafish infected with fluorescent M. abscessus R or S carrying tdTomato or Wasabi encoding plasmids[10]. From 10 dpi, we noted a heterogeneous mix of free bacteria around the peritoneal cavity or cellular granulomas diffuse foci of bacteria spread throughout host abdominal organ tissue, and necrotic granulomas with stereotypical host nuclei ringing around a central necrotic core containing M. abscessus in all animals (Fig. 2A).

We also observed the appearance of very large necrotic granulomas filled with fluorescent bacteria expressing tdTomato and necrotic debris measuring over 500 μm in M. abscessus R-infected animals from 14 dpi onwards (Fig. 2A). These large granulomas were observed only occasionally and at a rate of no more than 1 per infected animal at 14 and 28 dpi ($n = 2$ with single abscess, 9 without abscess). Three M. abscessus R-infected animals were maintained until 70 dpi and appeared to be outwardly healthy. All three were found to have multiple necrotic granulomas containing fluorescent M. abscessus R, demonstrating that persistent infection is possible in adult zebrafish, and two

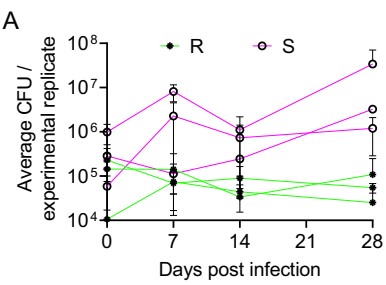
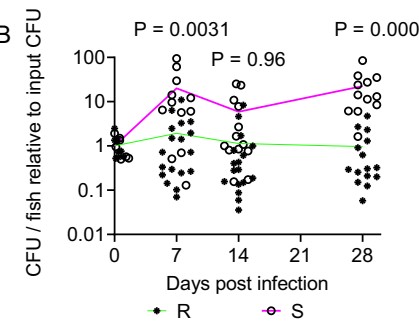

**Fig. 1** *M. abscessus* **establishes chronic infection in adult zebrafish. A** Enumeration of CFUs from adult zebrafish infected with either the R or the S variant of *M. abscessus*. Each point represents a single experimental replicate with at least three animals per timepoint. Total n per timepoint: 0 dpi R = 13 S = 12; 7 dpi R = 18 S = 12; 14 dpi R = 17 S = 13; 28 dpi R = 15 S = 12. **B** Relative CFUs recovered from adult zebrafish infected with either the R or the S variant of *M. abscessus*. Absolute CFU values were normalised to the inoculum CFU for each experimental replicate. Data are pooled from three replicates per *M. abscessus* variant. Total n per timepoint: 0 dpi R = 13 S = 12; 7 dpi R = 18 S = 12; 14 dpi R = 17 S = 13; 28 dpi R = 15 S = 12. Statistical tests by two-sided Student's *t* test at each timepoint. Data are presented as mean values ± SD. Source data are provided as a Source Data file.

were found to have necrotic granulomas measuring over 500 μm suggesting an increase in the rate of large granuloma formation as infections progress (Supplemental Fig. 1). Granulomas in *M. abscessus* S-infected animals did not reach this size at the 14 and 28 dpi timepoints sampled (n = 17 at 14 dpi, 16 at 28 dpi without abscess).

We next quantified the number of necrotic granulomas and "unorganised lesions", consisting of either diffuse foci of bacteria spread throughout host tissue or free bacteria around the peritoneal cavity. The proportion of necrotic granulomas in *M. abscessus* R-infected adult zebrafish increased between 10 and 14 dpi then remained similar between 14 and 28 dpi (Fig. 2B). *M. abscessus* S was observed to grow freely in mesenteric spaces and the rate of necrotic granulomas increased across all times with a statistically significant difference between 10 and 28 dpi (Fig. 2B). The proportion of necrotic granulomas was higher in *M. abscessus* R-infected than in *M. abscessus* S-infected animals at 14 dpi but similar at 10 and 28 dpi, suggesting granuloma formation is accelerated in R infections compared to S infections.

These patterns were recapitulated in our quantification of fluorescent bacterial burden in each type of lesion per animal. Significantly more *M. abscessus* R was observed within necrotic granulomas at 14 and 28 dpi than at 10 dpi, and an increase in the proportion of *M. abscessus* S within necrotic granulomas was only observed at 28 dpi compared to 10 dpi (Fig. 2C). The proportion of *M. abscessus* R within necrotic granulomas was higher than the proportion of *M. abscessus* S within necrotic granulomas at 14 dpi but similar at 10 and 28 dpi, again suggesting accelerated granuloma formation in R variant infections compared to S.

**Tumour necrosis factor-mediated immunity differentially controls infection by *M. abscessus* variants.** The cytokine TNF is essential for the granulomatous containment of *M. abscessus* in zebrafish embryos[23]. To visualize *tnfa* transcription, we next took advantage of the *TgBAC(tnfa:GFP)pd1028* zebrafish line[31], where GFP expression is driven by the *tnfa* promoter, to investigate if TNF expression is linked to granuloma formation. Expression of GFP was observed in adult zebrafish infected with either variant of *M. abscessus*. GFP was expressed by host cells in close contact with either *M. abscessus* R (Fig. 3A) or S (Fig. 3B).

To examine the role of host inflammatory signalling in the adult zebrafish-*M. abscessus* infection model, we first treated infected adults with the potent anti-inflammatory drug dexamethasone[32]. Dexamethasone treatment reduced *tnfa* promoter activity around *M. abscessus* lesions in *TgBAC(tnfa:GFP)pd1028* zebrafish (Fig. 3C). Additionally, and as expected, dexamethasone treatment increased

both R and S *M. abscessus* burdens (Fig. 3D), indicating that ablation of the inflammatory response led to uncontrolled bacterial expansion.

As dexamethasone acts broadly, we next specifically examined the role of the TNF-TNFR axis by knocking down expression of *tnfa* using a pooled gRNA CRISPR-Cas9 strategy. Comparison of WT and scrambled gRNA/Cas9-injected clutchmates indicated no effect of scrambled gRNA/Cas9-injection on *M. abscessus* R or S burdens (Supplemental Fig. 2). The efficacy of knockdown was validated by reduced RT-qPCR detection of *tnfa* transcripts in CRISPR-Cas9 injected adults (Fig. 4A). Knockdown of *tnfa* reduced the burden of R *M. abscessus* and increased the burden of S *M. abscessus* at 14 dpi (Fig. 4B).

While *tnfa*-depleted animals formed necrotic granulomas at the same rate as control animals in response to R *M. abscessus* infection (Fig. 4C), there was a sharply reduced proportion of *M. abscessus* R variant found within necrotic granulomas in *tnfa*-depleted animals (Fig. 4D). These parameters were unchanged in S *M. abscessus* infections suggesting the increased burden in S-infected *tnfa* knockdown animals was due to non-granulomatous mechanisms.

To further investigate the role of TNF signalling in our *M. abscessus* infection model, we next individually targeted the genes encoding the two zebrafish TNF receptors *tnfr1* and *tnfr2* for knockdown using CRISPR-Cas9 injections. Knockdown was confirmed by high resolution melt analysis of amplicons containing the first CRISPR target site demonstrating efficient editing of the loci (Fig. 4E). Knockdown of *tnfr1* or *tnfr2* again reduced the burden of R *M. abscessus* and increased the burden of S *M. abscessus* at 14 dpi (Fig. 4F).

Necrotic granuloma formation was reduced in both *tnfr1* and *tnfr2* knockdown animals infected with R *M. abscessus*, but unaffected in S *M. abscessus*-infected animals (Fig. 4G). The proportion of R *M. abscessus*, but not S *M. abscessus*, found within necrotic granulomas was also reduced in both *tnfr1* and *tnfr2* knockdown animals (Fig. 4H).

These observations demonstrate conservation of protective TNF-mediated immunity against the S variant in adult infections and uncover a detrimental role for the TNF-TNFR axis in promoting R variant immunopathology through granuloma necrosis.

**T cell-mediated immunity differentially controls infection by *M. abscessus* variants.** Given the requirement for T cells to maintain granuloma structure in adult zebrafish *M. marinum* infection[29], we next asked if there was T cell involvement around *M. abscessus* granulomas using *TgBAC(lck:EGFP)vcc4* zebrafish[33]. We observed T cell association and penetration throughout cellular

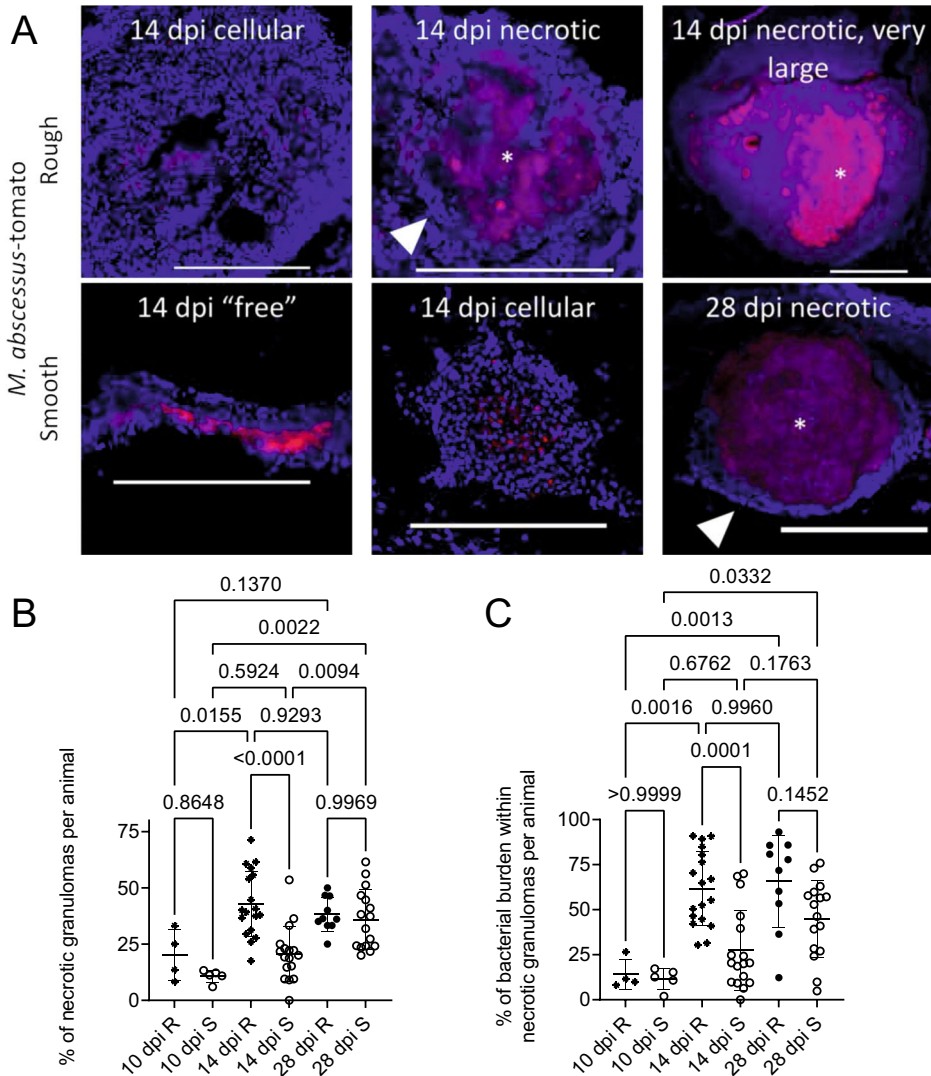

**Fig. 2 Granuloma histopathology is accelerated during *M. abscessus* R infection compared to S. A** Stereotypical examples of bacterial lesions from DAPI-stained sections from adult zebrafish infected with *M. abscessus* expressing tdTomato. Top row infected with the rough variant, bottom row infected with the smooth variant, timepoints as indicated. Images are representative of the two experimental replicates quantified in **B**, **C**. Scale bars indicate 200 μm. Filled arrowheads indicate epithelised macrophage nuclei forming a stereotypical concentric layer surrounding the mycobacterial core of necrotic granulomas, * indicate necrotic cores. **B** Quantification of bacterial lesion necrosis in adult zebrafish infected with approximately $10^5$ CFU *M. abscessus*. Each data point represents the proportion of lesions from a single animal. **C** Quantification of bacterial burden stratified by lesion necrosis in adult zebrafish infected with either the R or the S variant of *M. abscessus*. Each data point represents the proportion of lesions from a single animal. Total individual lesions and animals analysed in B and C pooled from two experimental replicates (necrotic/unorganised/animals): 10 dpi R (37/228/4); 10 dpi S (107/788/5); 14 dpi R (299/447/18); 14 dpi S (111/539/17); 28 dpi R (149/217/10); 28 dpi S (382/603/16). Statistical testing by two-sided ANOVA. Data are presented as mean values ± SD. Source data are provided as a Source Data file.

and necrotic *M. abscessus* R granulomas (Fig. 5A). We did not observe T cell interaction with *M. abscessus* S growing "freely" around peritoneal organs early in infection (Fig. 5B) and the T cell response to tissue-invasive *M. abscessus* S was noticeably less than that for equivalent sized *M. abscessus* R granulomas (Fig. 5C).

To directly test the requirement of T cells for containing *M. abscessus*, we next utilised the T cell-deficient *lck*$^{-/- sa410}$ mutant line. We infected wild type (WT) control and *lck*$^{-/- sa410}$ mutant adult zebrafish with both the S and R variants. T cell-deficient adult zebrafish were significantly more susceptible to *M. abscessus* R infection with reduced survival over 28 days of infection ($P = 0.0005$, Log-rank test) (Fig. 5D). T cell deficiency had a less pronounced effect on the survival of animals infected with *M. abscessus* S compared to *M. abscessus* R infection (WT S versus lck$^{-/-}$ S; $P = 0.03$, Log-rank test). Within the T cell-deficient animals, there

was a 5.5 day increased median survival for *M. abscessus* S-infected animals (34 dpi) compared to *M. abscessus* R (28.5 dpi), although both groups eventually succumbed to infection at the same rate after 35 dpi ($P = 0.78$, Log-rank test). Bacterial burden was significantly increased at 14 dpi in *lck*$^{-/- sa410}$ animals infected with the R, but not the S variant compared to burdens in WT adult zebrafish (Fig. 5E).

Surprisingly, we found necrotic granulomas in a survivor 56 dpi *lck*$^{-/- sa410}$ fish infected with *M. abscessus* R (Supplemental Fig. 3). These granulomas were all relatively small, with the largest having necrotic cores of approximately 100 μm, compared to the large granulomas seen in our 70 dpi WT animals at more than 500 μm (Supplemental Fig. 1).

These observations demonstrate that the control of *M. abscessus* R infection is more reliant on T cell-mediated immunity

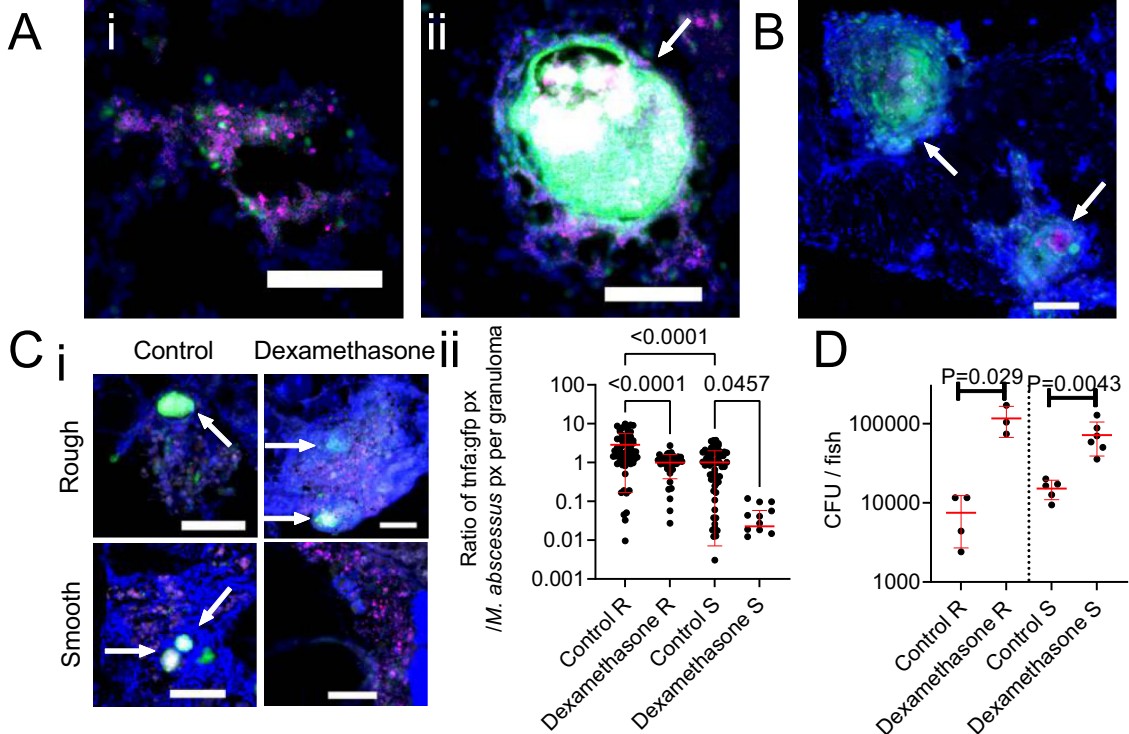

**Fig. 3 Host *tnfa* expression is driven by *M. abscessus* infection and control of *M. abscessus* infection is sensitive to immunosuppression by dexamethasone. A** Examples of R variant *M. abscessus*-tdTomato lesions in DAPI-stained cryosections from a 14 dpi *TgBAC(tnfa:GFP)*^pd1028^ adult zebrafish. i. Cellular lesion, ii. Necrotic granuloma. Arrow indicates necrotic granuloma, *M. abscessus*-tdTomato is coloured magenta, *tnfa* promoter induction is marked in green. Scale bars indicate 100 μm. Images are representative of two biological replicates of 3 animals per replicate. **B** Examples of S variant *M. abscessus*-tdTomato lesions in DAPI-stained cryosections from a 28 dpi *TgBAC(tnfa:GFP)*^pd1028^ adult zebrafish. Arrows indicate necrotic granulomas, *M. abscessus*-tdTomato is coloured magenta, *tnfa* promoter induction is marked in green. Scale bar indicates 100 μm. Images are representative of two biological replicates of 3 animals per replicate. **C** i. Examples of *tnfa* promoter activity in 14 dpi *TgBAC(tnfa:GFP)*^pd1028^ adult zebrafish infected with *M. abscessus*-tdTomato and treated with dexamethasone as indicated. Arrows indicate necrotic granulomas, *M. abscessus*-tdTomato is coloured magenta, *tnfa* promoter induction is marked in green. Scale bars indicate 100 μm. Control images are representative of 3 animals per *M. abscessus* variant, dexamethasone images are representative of 2 animals per *M. abscessus* variant. ii. Quantification of GFP pixels as a function of *M. abscessus*-tdTomato fluorescence in *TgBAC(tnfa:GFP)*^pd1028^ adult zebrafish. Data are a single experimental replicate, each point represents a single granuloma, total animals per group: Control R = 3, Dexamethasone R = 2, Control S = 3, Dexamethasone S = 3. Statistical testing by two-sided ANOVA. **D** Enumeration of CFUs from adult zebrafish infected with either the R or the S variant of *M. abscessus* and treated with dexamethasone. Each point represents a single animal and data are representative of 2 biological replicates. Total number of animals: Control R = 4, Dexamethasone R = 3, Control S = 5, Dexamethasone S = 6. Statistical testing by two-sided Student's *t* test within each of the independent R and S experiments. Data are presented as mean values ± SD. Source data are provided as a Source Data file.

than *M. abscessus* S infection during the initial 2 weeks of infection.

**Regulatory T cells interact with rough *M. abscessus* granulomas to control infection burden.** To further investigate the role of host immunopathology in *M. abscessus* infection, we next investigated the recruitment of regulatory T cells (Tregs) to sites of *M. abscessus* infection using the *TgBAC(foxp3a:TagRFP,-cryaa:EGFP)*^vcc3^ transgenic line where Tregs are labelled with RFP expression[33]. Consistent with the recruitment profile of *TgBAC(lck:EGFP)*^vcc4^ positive T cells, we observed Treg association and penetration throughout cellular and necrotic *M. abscessus* R granulomas from 14 dpi (Fig. 6A), and to necrotic *M. abscessus* S granulomas from 28 dpi (Fig. 6B). The Treg response to *M. abscessus* R granulomas was significantly higher than for equivalent sized *M. abscessus* S granulomas at 14 dpi (Fig. 6C). Although we observed a higher Treg response to *M. abscessus* S than R at 28 dpi (Fig. 6C), this was driven by the significantly higher burden of *M. abscessus* R than S per granuloma rather than an absolute increase in Treg fluorescence per granuloma (28 dpi median *M. abscessus* fluorescence R: 2072, *n* = 66; S: 41.00,

*n* = 29; Mann–Whitney U test *P* < 0.0001. Median Treg fluorescence R: 1298, *n* = 66; S: 123.0, *n* = 29; Mann–Whitney U test *P* < 0.0001).

As we had only observed significant Treg involvement with *M. abscessus* R granulomas, we next asked if hyperinflammatory *foxp3a*^vcc6^ mutant zebrafish would have exacerbated *M. abscessus* R-infection. Homozygous mutant *foxp3a*^vcc6/vcc6^ zebrafish had a higher infection burden than WT clutchmate controls (Fig. 6D). However, there were similar percentages of necrotic granulomas and similar burdens of *M. abscessus* within necrotic granulomas for each group at 14 dpi (Fig. 6E, F).

## Discussion

In this study, we report the use of adult zebrafish to probe both host and mycobacterial determinants of pathogenesis during persistent infection with *M. abscessus*. Infection with the R and S variants was maintained at high levels up to one month post infection in genetically intact animals, a major improvement on traditional mouse models of *M. abscessus* infection.

It is well known that the intracellular lifestyle of the R and S morphotypes differ significantly, resulting in entirely distinct

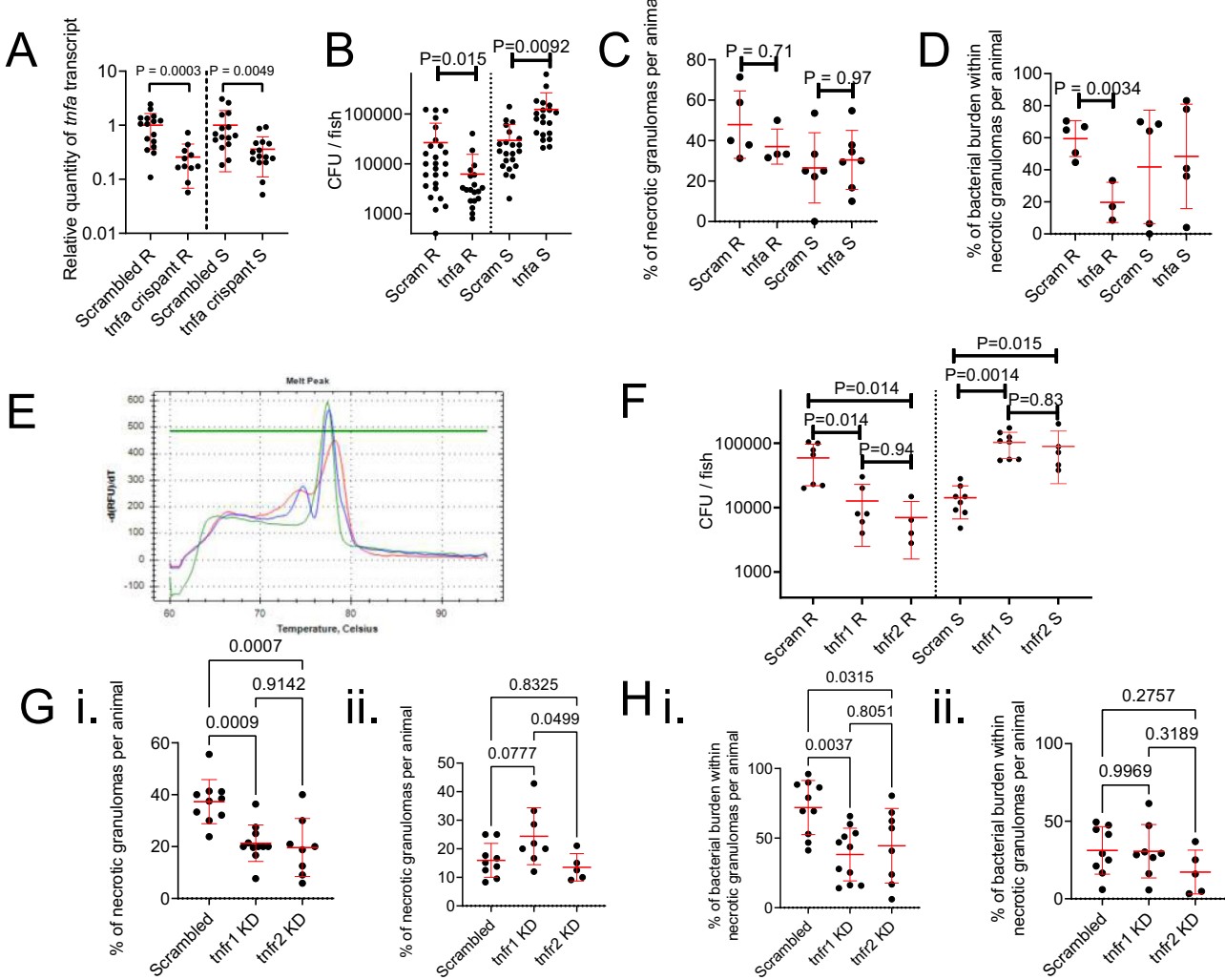

**Fig. 4 Host TNF drives *M. abscessus* R granuloma expansion and control of *M. abscessus* S infection. A** Quantification of *tnfa* transcripts activity in 14 dpi zebrafish injected with scrambled control or *tnfa*-targeting gRNA-Cas9 complexes. Each data point represents a single animal, data are pooled pooled from 2 biological replicates, total number of animals: Scrambled R = 16, tnfa crispant R = 11, Scrambled S = 15, tnfa crispant S = 15. Statistical testing by two-sided Student's *t* test within each of the independent R and S experiments. **B** Enumeration of CFUs from *tnfa* crispant adult zebrafish infected with $10^4$ CFU of either the R or the S variant of *M. abscessus*. Each data point represents a single animal, data are pooled from 3 biological replicates, total number of animals: Scrambled R = 26, tnfa crispant R = 19, Scrambled S = 22, tnfa crispant S = 20. Statistical testing by two-sided Student's *t* test within each of the independent R and S experiments. **C** Quantification of bacterial lesion necrosis in *tnfa* crispant adult zebrafish infected with *M. abscessus*. Each data point represents the proportion of lesions from a single animal. Statistical testing by two-sided ANOVA. **D** Quantification of bacterial burden stratified by lesion necrosis in *tnfa* crispant adult zebrafish infected with *M. abscessus*. Each data point represents the proportion of lesions from a single animal. Total individual lesions and animals analysed in **C**, **D** (necrotic/unorganised/animals): Scrambled R (43/51/5); *tnfa* crispant R (12/21/3); Scrambled S (25/43/5); *tnfa* crispant S (57/96/5). Statistical testing by two-sided ANOVA. **E** Example of high resolution melt analysis of the first *tnfr2* CRISPR target site in 14 dpi zebrafish injected with scrambled control (blue), *tnfr1* (green), or *tnfr2* (red)-targeting gRNA-Cas9 complexes. **F** Enumeration of CFUs from *tnfr1* and *tnfr2* crispant adult zebrafish infected with $10^4$ CFU of either the R or the S variant of *M. abscessus*. Each data point represents a single animal, data are representative of 3 biological replicates. Statistical testing by two-sided ANOVA within each of the independent R and S experiments. Total number of animals (Scram/tnfr1/tnfr2): R = 7/6/4, S = 8/8/5. **G** Quantification of bacterial lesion necrosis in *tnfr1* and *tnfr2* crispant adult zebrafish infected with *M. abscessus*. Each data point represents the proportion of lesions from a single animal. **H.** Quantification of bacterial burden stratified by lesion necrosis in *tnfr1* and *tnfr2* crispant adult zebrafish infected with *M. abscessus*. Each data point represents the proportion of lesions from a single animal. Total individual lesions and animals analysed in **G**, **H** (necrotic/unorganised/animals): Scrambled R (91/224/10); *tnfr1* crispant R (89/308/11); *tnfr2* crispant R (40/173/8); Scrambled S (25/139/9); *tnfr1* crispant S (40/143/8); *tnfr2* crispant S (17/97/5). Statistical testing by two-sided ANOVA. Data are presented as mean values ± SD. Source data are provided as a Source Data file.

infection scenarios that, we hypothesise, underlie the accelerated granuloma formation by the R variant in adult zebrafish[10,34]. The absence of GPL on the outer mycobacterial membrane causes corded growth of R variants, resulting in multiple bacilli being simultaneously phagocytosed by macrophages and overloaded phagosomes that rapidly activate autophagy pathways[12,34]. Comparatively, the S variant is able to survive for an extended

period of time within the phagosome, producing a chronic and persistent infection[10,15]. As such, these polar infection responses may explain why the R variant displays widespread necrotic granuloma formation by 14 dpi, compared to the S variant that shows delayed onset of granuloma formation after 14 dpi. Moreover, this observation matches the superior in vivo growth performance of S bacilli compared to R, suggesting that the R

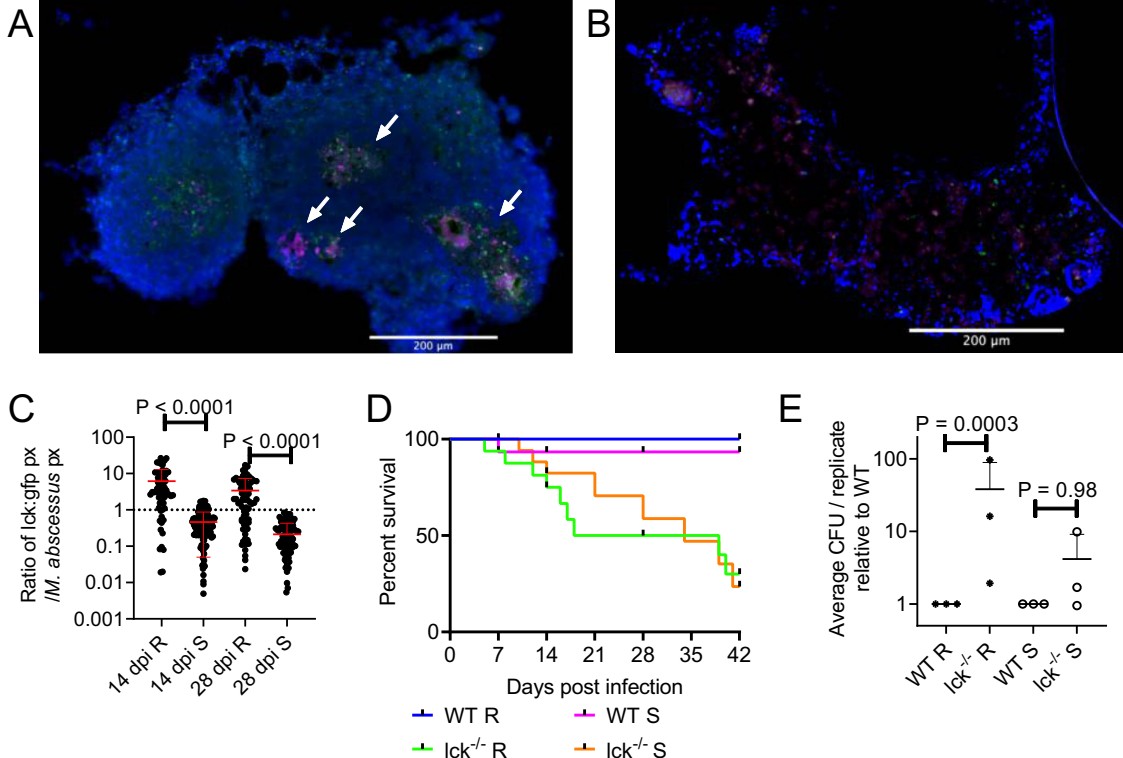

**Fig. 5 T cell-mediated immunity differentially controls infection by *M. abscessus* variants. A** Examples of T cell recruitment to granulomas in 14 dpi *TgBAC(lck:EGFP)^{vcc4}* adult zebrafish infected with R *M. abscessus*-tdTomato. **B** Example of lack of T cell recruitment to S *M. abscessus*-tdTomato in a large cellular lesion from a 14 dpi *TgBAC(lck:EGFP)^{vcc4}* adult zebrafish. Scale bars indicate 200 µm. Arrows indicate necrotic granulomas, *M. abscessus*-tdTomato is coloured magenta, *lck:gfp* positive T cells are marked in green. Images from **A** and **B** are representative of the dataset analysed in **C**. **C** Quantification of T cell GFP pixels as a function of *M. abscessus*-tdTomato fluorescence in *TgBAC(lck:EGFP)^{vcc4}* adult zebrafish. Each point represents a single lesion. Total n per group (lesions/animals): 14 dpi R = 74/2, S = 137/3; 28 dpi R = 91/3, S = 117/2. **D** Survival analysis of WT and lck$^{-/-}$ $^{sa410}$ adult zebrafish infected with R or S *M. abscessus*. Total $n$ = 12 WT/*Mabs* R; 16 lck$^{-/-}$/*Mabs* R; 15 WT/*Mabs* S; 22 lck$^{-/-}$/*Mabs* S. **E** Normalised CFUs recovered from 14 dpi WT and lck$^{-/-}$ $^{sa410}$ adult zebrafish infected with *M. abscessus*. Each point represents the average of a single experiment with at least 2 animals per group. Total $n$ per group: R WT = 12 lck$^{-/-}$ = 15; S WT = 11 lck$^{-/-}$ = 12. Statistical testing by two-sided ANOVA. Data are presented as mean values ± SD. Source data are provided as a Source Data file.

variant is at an overall disadvantage because of its intrinsic hyper-inflammatory status and the activation of T cell-mediated immunity that occurs concomitantly with granuloma formation. Interestingly, earlier reports using the zebrafish embryo demonstrated that both bacterial burden and granuloma formation dynamics were similar between both the S and R variants[10,26], highlighting the critical role of adaptive immunity in divergent responses to the *M. abscessus* variants. Taken together, our data provide additional evidence for the distinct intracellular fates of both S and R variants in vivo, and further implicate the role of adaptive immunity in granuloma formation and the control of *M. abscessus* infection in adult zebrafish.

While dexamethasone-sensitive immunity was necessary for the control of both variants and Treg immunomodulation was necessary for the control of R infection, we discovered a striking pathogenic role for *tnfa, tnfr1*, and *tnfr2* during R infections and a protective role during S infections. Although dexamethasone treatment marginally reduced host *tnfa* promoter activity around sites of infection, dexamethasone treatment has wider immuno-suppressive effects. Our data points towards a specific role for host TNF signalling being co-opted by *M. abscessus* R infection to drive to the formation of necrotic granulomas as a permissive niche for bacterial growth within the host. The detrimental role of *tnfa, tnfr1*, and *tnfr2* in adult zebrafish-*M. abscessus* R infection is the opposite of previously published work in the zebrafish embryo using knockdown of *tnfr1*[23], which points to the

importance of using multiple models systems to investigate different stages of infection. Our data suggest human studies of increased susceptibility to *M. abscessus* in patients receiving TNF inhibition should be stratified by strain morphology to investigate the clinical relevance of this effect. We hypothesise that the increased susceptibility of patients receiving TNF inhibition therapy to *M. abscessus* disease will be driven by the loss of control of S variant infections[35].

Our finding that Tregs are closely associated with *M. abscessus* R granulomas and act to restrain *M. abscessus* R growth provides complementary evidence that *M. abscessus* R takes advantage of host immunopathogenesis to establish chronic infection in adult zebrafish. While the assays proved adept at detecting reduced necrosis in the *tnfa, tnfr1*, and *tnfr2* crispants, Treg-deficient mutants had comparable proportions of necrotic granulomas and proportions of *M. abscessus* R within necrotic granulomas to WT clutchmates indicating no increase in necrotic granuloma phenotypes. We hypothesise that this is because our granuloma necrosis assays for *M. abscessus* R are running at the upper limits of signal. Zebrafish Treg cells produce context-specific regulatory factors in a Foxp3a-dependent manner[36], and future experiments are warranted to identify the protective Foxp3a-dependent factors produced in the context of *M. abscessus* R granulomas.

T cells are critical host determinants in the control of myco-bacterial infection[37]. Recruitment of T cells into granulomas is thought to be essential in containing persistent infection, while T

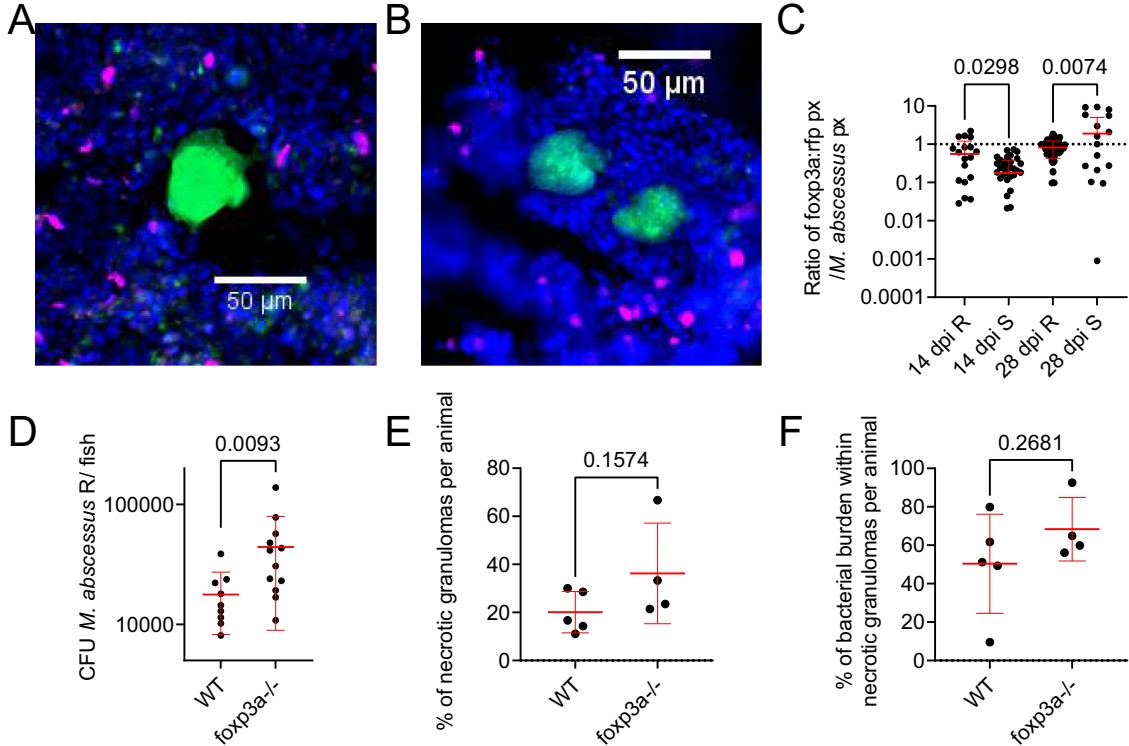

**Fig. 6 Regulatory T cells interact with rough *M. abscessus* granulomas to control infection burden. A** Example of Treg recruitment to a necrotic R *M. abscessus*- wasabi granuloma in 14 dpi *TgBAC(foxp3a:TagRFP,cryaa:EGFP)$^{vcc3}$* adult zebrafish. **B** Example of Treg recruitment to a necrotic S *M. abscessus*-wasabi granuloma from a 28 dpi *TgBAC(foxp3a:TagRFP,cryaa:EGFP)$^{vcc3}$* adult zebrafish. Scale bars indicate 50 μm. *M. abscessus*-wasabi is coloured green, *foxp3a:TagRFP* positive Treg are marked in magenta. Images from A and B are representative of the dataset analysed in **C**. **C** Quantification of Treg RFP pixels as a function of *M. abscessus*-wasabi fluorescence in *TgBAC(foxp3a:TagRFP,cryaa:EGFP)$^{vcc3}$* adult zebrafish. Each point represents a single lesion. Total n per group (lesions/animals): 14 dpi R = 26/4, S = 55/3; 28 dpi R = 66/3, S = 29/3. Statistical testing by 2-way ANOVA. **D** Enumeration of CFUs from *foxp3a$^{vcc6}$* mutant adult zebrafish infected with $2 \times 10^4$ CFU *M. abscessus* R. Each data point represents a single animal, data are representative of two biological replicates. Statistical testing by two-sided T test. Total number of animals: WT = 9, foxp3a−/− = 12. **E** Quantification of bacterial lesion necrosis in *foxp3a$^{vcc6}$* mutant adult zebrafish infected with *M. abscessus* R. Each data point represents the proportion of lesions from a single animal. **F** Quantification of bacterial burden stratified by lesion necrosis in *foxp3a$^{vcc6}$* mutant adult zebrafish infected with *M. abscessus* R. Each data point represents the proportion of lesions from a single animal. Total individual lesions and animals analysed in **E**, **F** (necrotic/unorganised/animals): WT (35/91/5); *foxp3a$^{vcc6}$* mutant (22/97/4). Statistical testing by two-sided T test. Data are presented as mean values ± SD. Source data are provided as a Source Data file.

cell deficiencies are associated with greater mycobacterial infection severities[27,37–39]. Recently, an adult zebrafish infection model for *M. leprae* demonstrated that T cells are essential for containment of infection[27]. Herein, we examined the recruitment of T cells within granulomas and identified that S variant granuloma were characterised by a relative paucity of T cell infiltration, suggesting that T cells play a less significant role in S variant infections. Consistent with this, we observed increased in vivo growth of the R, but not S, variant in the absence of T cells when compared to WT animals and in our co-infection experiments where R but not S burdens were increased in *lck$^{−/−}$* zebrafish. Following the hypothesis that patients acquire an environmental S variant of *M. abscessus*, our findings provide insight into the clinical observation that AIDS patients are not at increased risk of *M. abscessus* infection to the same degree that AIDS is a risk factor for *M. tuberculosis* and other non-tuberculous mycobacterium infections, such as *Mycobacterium avium*, as T cell deficiency has a limited effect on the control of S variant infection[37].

The extended maintenance of R variant burden for at least 4 weeks in zebrafish is comparable to our recent data from the C3HeB/FeJ mouse[40], but the proliferation of the S variant up to a log above inoculation dose is unprecedented in a genetically intact

vertebrate host. The granulomatous immunopathology in mycobacterium-infected C3HeB/FeJ mice is due to an exaggerated type I interferon response suppressing protective IL-1[41]. Further analysis of IFN and IL-1 responses to *M. abscessus* infection of mice and zebrafish will help translate our understanding of these dichotomous responses into host directed therapies.

We did not observe switching of S *M. abscessus* into a rough colony morphotype at any timepoint during this study. In vivo switching is a rare event that has only been documented in immunocompromised mice or after months-to-years of infection in patients[13,42]. The high S morphotype burdens achieved in adult zebrafish suggest this platform may be useful for future studies of switching during extended infections, with the potential to model responses to chemotherapy and transcriptional adaptations to host residence that may include rough-like surface morphologies[13,43].

To date, our understanding of the diverse immune responses between S and R variants have essentially been thoroughly described with respect to innate immunity, and currently our knowledge pertaining to adaptive immunity in *M. abscessus* infection has been poorly characterised[20]. Using this new adult zebrafish *M. abscessus* infection model, we have shown that S and

R variants produce strikingly different disease phenotypes, driven by unique interactions with host immunity. Consequently, these results suggest that the host-pathogen interactions dictating *M. abscessus* pathogenesis are complex and implicate different arms of host immunity to a greater extent than originally anticipated. Future work should exploit this relevant animal model in combination with zebrafish lacking the CFTR gene, and for the development and testing of novel antibiotics and vaccine candidates that may be used for the treatment of *M. abscessus* infection. The robust infection phenotypes created by exemplar R and S *M. abscessus* strains suggest adult zebrafish will be an important platform to study *M. abscessus* virulence factors and clinical strain genetic variation.

## Methods

**Zebrafish strains and handling**. Zebrafish strains used in this study were AB strain wildtype, *TgBAC(tnfa:GFP)^pd1028*, *TgBAC(lck:EGFP)^vcc4*, *lck^−/− sa410* mutant, *TgBAC(foxp3a:TagRFP,cryaa:EGFP)^vcc3*, *foxp3a^vcc6* mutant[31,33] between 3 and 6 months of age. Animals were held in a 28 °C incubator with a 14:10 h light:dark cycle. Animals were infected by intraperitoneal injection with approximately 10⁵ CFU *M. abscessus*, unless otherwise stated, using a 31 G insulin needle and syringe, as previously described[44]. Initial inoculum dose was confirmed by plating of inoculum dose and homogenisation of injected fish immediately after infection. Infected zebrafish were recovered into system water (approximately 750 µs conductivity, pH 7.4) and held in 1 L beakers at a density of approximately one fish per 50–100 ml water. Animals were cleaned daily with a 20% water change and fed once per day for the duration of experiments as previously described[44]. Infection experiments were carried out with ethical approval from the Sydney Local Health District Animal Welfare Committee approval 16-037.

The vcc allele zebrafish lines are available on request from the corresponding author (Sydney, Australia or Singapore) or the creator Dr Kazu Kikuchi (current address Osaka, Japan). The *lck^−/− sa410* zebrafish line is a publicly available resource from the Zebrafish International Resource Center (Oregon, USA). The *TgBAC(tnfa:GFP)^pd1028* zebrafish line is available from the corresponding author or the creators at Duke University School of Medicine (North Carolina, USA).

**M. abscessus strains and handling**. Rough (R) and smooth (S) variants of *M. abscessus* strain CIP104536^T were grown at 37 °C in Middlebrook 7H9 broth supplemented with 10% Oleic acid/Albumin/Dextrose/Catalase (OADC) enrichment and 0.05% Tween 80 or on Middlebrook 7H10 agar containing 10% OADC (7H10 OADC). Recombinant *M. abscessus* strains expressing tdTomato or Wasabi were grown in the presence of 500 µg/ml hygromycin[10,23]. Homogenous bacterial suspensions for intraperitoneal injection in adult fish were prepared as previously reported[45]. Briefly, *M. abscessus* was plated from freezer stocks onto 7H10 OADC supplemented with 500 µg/ml hygromycin. Colonies were picked and outgrown in 7H9 OADC 500 µg/ml hygromycin for three days. *M. abscessus* was harvested by pelleting, resuspended in 7H9 and sheered through at 27 G needle prior to aliquoting and freezing.

The fluorescent *M. abscessus* strains are available from the corresponding author or their creator Dr Laurent Kremer at IRIM (Montpellier, France).

**Bacterial recovery**. Animals were killed by tricaine anaesthetic overdose (>300 µg/ml) and rinsed in sterile water. Individual carcasses were mechanically homogenised and serially diluted into sterile water. Homogenates were plated onto 7H10 supplemented with OADC and 300 µg/ml hygromycin. Plates were grown for at least 4 days at 37 °C.

**Histology**. Animals were subjected to cryosectioning as previously described[44]. Briefly, euthanasia was performed by tricaine anaesthetic overdose and specimens were fixed for 2–4 days in 10% neutral buffered formalin at 4 °C. Specimens were then rinsed in PBS, incubated overnight in 30% sucrose, incubated overnight in 50/50 30% sucrose and Tissue-Tek O.C.T. compound (OCT, Sakura Finetek), and finally incubated overnight in OCT prior to freezing at −80 °C. Cryosectioning was performed to produce 20 µm thick sections. Sections were post-fixed in 10% neutral buffered formalin and rinsed in PBS prior to further processing. Slides for fluorescent imaging were mounted with coverslips using Fluoromount G containing DAPI (Proscitech).

T cells were detected in sections from *TgBAC(lck:EGFP)^vcc4* zebrafish by anti-GFP staining to enhance visible fluorescent green signal (primary antibody: ab13970, Abcam; secondary antibody: ab150173, Abcam), both antibodies were diluted 1:500 in 5% goat serum (Thermofisher). Stained slides were then mounted with coverslips using Fluoromount G containing DAPI. All imaging was carried out on a Leica DM6000B microscope with LAS software version X (Leica) and image analysis was performed in ImageJ version 1.53n (NIH).

**Drug treatment**. Animals for dexamethasone treatment were infected with 1–5 × 10⁴ CFU to ensure survival following immunosuppression. Water soluble dexamethasone (Sigma, D2915) was added to a final concentration of 40 µg/ml immediately following infection and changed every second day.

**CRISPR-Cas9 gene knockdown**. Embryos were injected at the single cell stage with injection mix containing 1 µl phenol red, 2 µl 500 ng/µl pooled guides, and 2 µl 10 µM Cas9. Control embryos are injected with scrambled guide RNA as previously described[46]. The oligo sequences used to generate *tnfa*-targeting guides were computed by Wu et al.[46]: guide 1 TAATACGACTCACTATAGGTTGAGAGTCG GGCGTTTTGTTTTAGAGCTAGAAATAGC, guide 2 TAATACGACTCACTAT AGGTCTGCTTCACGCTCCATAGTTTTAGAGCTAGAAATAGC, guide 3 TAA TACGACTCACTATAGGGATTATCATTCCCGATGAGTTTTAGAGCTAGAA ATAGC, guide 4 TAATACGACTCACTATAGGTCCTGCGTGCAGATTGAGGT TTTAGAGCTAGAAATAGC. Oligos for *tnfr1*-targeting guides: guide 1 TAATA CGACTCACTATAGGGTGATGAAGGAACCTACCGTTTTAGAGCTAGAAAT AGC, guide 2 TAATACGACTCACTATAGGGTGAAGACTCCCTGGAATGTTT TAGAGCTAGAAATAGC, guide 3 TAATACGACTCACTATAGGCATTTTTA TGGCATACGAGTTTTAGAGCTAGAAATAGC, guide 4 TAATACGACTCACT ATAGGTCTTTAAATCGTCCGATAGTTTTAGAGCTAGAAATAGC. Oligos for *tnfr2*-targeting guides guide 1 TAATACGACTCACTATAGGTCTGAGTACATTC CATCTGTTTTAGAGCTAGAAATAGC, guide 2 TAATACGACTCACTATAG GTTCTTTGAGCGCCTTCGCGTTTTAGAGCTAGAAATAGC, guide 3 TAATA CGACTCACTATAGGTGCCTCTTACTCGGTGTGGTTTTAGAGCTAGAAATA GC, guide 4 TAATACGACTCACTATAGGGTCATGTAGTCAGGCGGAGTTTT AGAGCTAGAAATAGC.:

Embryos were raised to 10 weeks post fertilisation and were then infected with a reduced burden of 1–5 × 10⁴ CFU, scaled to their smaller size.

Genotyping of *tnfr1* and *tnfr2* crispants was performed by high resolution melt analysis of amplicons produced by amplification of gDNA from diluted homogenate using oligos: *tnfr1* gFw GCAGTGCAGAAAACATGAGG, GRv CGTTTTGTGCATTGCTGGC; *tnfr2* gFw ACTCGCTTGTCTGTGCAATG, gRv TGGACACTTGAAACAATTGGGA. Amplification and analysis was carried out with MeltDoctor™ HRM Master Mix (Thermofisher) reagent on a CFX thermocycler with CFX Maestro software version 2.2 (Biorad) according to MeltDoctor product specifications.

Phenotyping of *tnfa* crispants was performed by RT-qPCR using oligos: *tnfa* qFw GCTTATGAGCCATGCAGTGA, qRv AAGTGCTGTGGTCGTGTCTG; 18 s qFw TCGCTAGTTGGCATCGTTTATG, qRv CGGAGGTTCGAAGACGATCA. Homogenate was processed for RNA extraction following product specification for Trizol LS (Thermofisher). Reverse transcription was carried out from 2 µg total RNA with the Applied Biosystems™ High Capacity cDNA Reverse Transcription Kit (Thermofisher). Quantitative PCR was carried out with PowerUp™ SYBR™ Green Master Mix (Thermofisher) on a CFX thermocycler with CFX Maestro software version 2.2 (Biorad).

**Statistics**. All statistical testing was carried out using Graphpad Prism version 9.3.1. Statistical tests are indicated in the figure legends and include both T tests for pairwise comparisons and ANOVA analyses with post hoc analysis for comparison of three or more groups. Each data point indicates a single animal unless otherwise stated.

**Reporting summary**. Further information on research design is available in the Nature Research Reporting Summary linked to this article.

## Data availability

The processed analysis data are included in the Source Data file that is provided with this paper. Raw image files will be archived for 10 years (2022–2032) by The Centenary Institute (Sydney, Australia). The raw image and analysis data are available under restricted access due to their large size and prohibitive cost of public hosting: they can be obtained upon request by email from the corresponding author, who will mediate access from the Centenary Institute servers. Expected response times will be within one week. Source data are provided with this paper.

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

## Acknowledgements

We thank the Centenary imaging facility core and Sydney Cytometry staff Drs Kristina Jahn, Angela Kurz, and David Liu, for their assistance. We thank Dr Renjing Liu for providing material supplies for this study. Australian National Health and Medical Research Council CJ Martin Early Career Fellowship APP1053407 and Project Grant APP1099912; The University of Sydney Fellowship G197581; NSW Ministry of Health under the NSW Health Early-Mid Career Fellowships Scheme H18/31086; the Kenyon Family Foundation Inflammation Award; Australian-French Association for Research and Innovation (AFRAN) Initiative; The University of Sydney Marie Bashir Institute 2019 Seed Funding to S.H.O. Sydney Medical School Summer Scholarship to J.Y.K. Post-doctoral fellowship granted by Labex EpiGenMed, an "Investissements d'avenir" program ANR-10-LABX-12-01 to M.D.J.; The Fondation pour la Recherche Médicale DEQ20150331719 to L.K.

## Author contributions

J.K. & K.W.: performed experiments; E.H. & E.K.: preliminary experiments; S.W., K.L., & T.C.: C.F.U. recovery assays; P.M.C.: histological analysis; K.K.: provided reagents; J.A.T. & W.J.B.: supervision of study; M.D.J.: conceived study, supervision of study, wrote manuscript; L.K.: conceived study, provided reagents, supervision of study, wrote manuscript; S.H.O.: conceived study, performed experiments, supervision of study, wrote manuscript.

## Competing interests

The authors declare no competing interests.
