## [Peer Review File · Nature Communications]

Rough and smooth variants of *Mycobacterium abscessus* are differentially controlled by host immunity during chronic infection of adult zebrafishREVIEWER COMMENTS

Reviewer #1 (Remarks to the Author):

I consider this as an interesting paper, which utilizes zebrafish model to study pathophysiology of smooth (S) and rough (R) variants of *M. abscessus*. This information is relevant as at the current stage, the knowledge of interaction between these different *M. abscessus* variants and immune cells in vivo is limited.

Earlier, zebrafish model has been used successfully to study other mycobacterial infection. The model is well chosen as it allows using various mutant lines and advanced microscopy. In addition, ethically this model is sound. In this current manuscript, methods are solid and well described. The paper is logically written and thus easy to follow. In conclusion, obtained results provide evidence for usage of zebrafish model for studying the *M. abscessus* infection.

General comments:

Although I do not have major criticism about the experiments, I think that technically the paper is somewhat limited. To make the paper stronger It would have been interesting to know if mechanisms regulating switch between R and S could be studied in this fish model.

Reviewer #2 (Remarks to the Author):

The authors report on an adult zebrafish model for *Mycobacterium abscessus*, a notoriously difficult to treat pathogen associated with high mortality in specific groups, such as those with cystic fibrosis. In humans infections, a stark difference has been observed between two morphotypes, rough and smooth based on the presence or absence of a surface glycopeptidolipid. The smooth morphotype possesses it and are more environmental and less invasive. The more virulent rough strain emerges during the course of infection in humans and is associated with more severe disease. The rough strain also survives in human monocytes in culture versus the smooth strain which is cleared, and the same is true for a lung infection model.

While the subject matter is interesting and important, this paper does not shed light on it. Unfortunately it presents a mishmash of poorly done, incorrectly analyzed experiments with no real interpretation. The first set of findings contradict the literature. The smooth strain appears to grow better in the zebrafish than the rough strain and forms more necrotic foci (Figure 1D versus 1C, even if the authors do not interpret it that way). This difference from the literature is not commented upon and rather the example granulomas presented in Figure 2 appear to contradict this results as the one with the most bacteria is the rough one (Figure 2A top, left panel). The oil red stains in panels B and C don't provide much information on the differences. The authors try to categorize lesions as organized and not organized in Figure 3 but this does not make much sense from the histology shown, and the comparisons shown of the quantitations do not either. In Figure 5, the authors find that animals succumb to both strains if they lack T cells, which is not surprising. However, they claim that "T cells are necessary to control the R strain but not the strain", a claim that is unsubstantiated by their findings. Their survival curve (panel A) in the T cell deficient (*lck*^{-/-}) animals are no different for the S and R, despite a barely significant p value provided by the authors. Their interpretation of panel B is also incorrect. Despite their finding a p value difference for the R and not the S strain, between wildtype and *lck*^{-/-} fish, the spread is large and the sample size small (3 fish). Also panel B is displayed in a strange way. One simply cannot conclude anything from this experiment. Even if true, what does the finding mean for pathogenesis. Similarly, in the next set of results, where they examine mixed infections, the authors claim that the increased growth of R is attenuated by co-infection with S. But this is from two animals and the spread in the S only is large. This needs to be repeated and if true, an interpretation of the results is needed.

In summary, this paper suffers from being technically flawed, with flawed statistical analyses, and no interpretation of the data with regard to how and why they differ from the literature and are internally contradictory as well.

Reviewer #3 (Remarks to the Author):

In this submission, the authors describe using a zebrafish model to examine differences in host immune response resulting from infection with different *Mycobacterium abscessus* variants. Their approach is thorough, evaluating both single and mixed infections, with appropriate endpoints of histology/fluorescence and bacterial culture. I found the paper to be generally well written. At times the results section strayed into discussion topics, but this fit the flow and may be acceptable for this journal. I do have a few minor comments as follows:

L227 - If S variant dropped, why is the ratio suggesting a greater proportion relative to R (0.5:1, R:S), when R had similar recoveries in single or mixed?

L265 - would the T cell depletion associated with AIDS not benefit *M. abscessus*? You saw greater burdens of R variant in T cell deficient zfish.

L340 - more detail is required for basic husbandry conditions. Such information is critical for reproducibility. What were the fish fed and how often? What are the water parameters (conductivity, pH)? Static tanks? Recirc? Flow through? Stocking density? How many animals held in each beaker following injection?

L359 - What dose of tricaine?

L360 - how was the homogenization done? Chemical or physical? Explain.

Reviewer #4 (Remarks to the Author):

This paper introduces the potential of adult zebrafish as a model of chronic *M. abscessus* infection. This is very interesting due to the lack of suitable models for studying the adaptive immune response to *M. abscessus*. The authors show differences in granuloma formation between smooth and rough *M. abscessus* morphotypes. They also show the importance of T cells in controlling rough *M. abscessus* infection, whereas the smooth morphotype appears to not induce the same T cell response. This very nicely brings together previous contradicting work regarding innate immune responses to smooth and rough morphotypes and clinical observations. Overall, I found the paper to be very interesting, showing novel work using adult zebrafish, and generally convincing. It is well written and clear to follow, with good referencing to the wider literature. I would recommend it be accepted for publication following some amendments (most importantly figures 1A and 1B).

Introduction - I would suggest making note of the idea that while an irreversible genetic switch can occur from smooth to rough, GPL production can also be controlled on a variable transcriptional scale between the smooth and rough morphotypes.

Figures 1A and 1B: I have the following comments

- I wonder if these would be better presented as box plots pooling all animals from across experimental repeats, rather than line graphs.
- I don't see the value in normalising the quantification data to the zero hour time points in Figure 1B. Changes in bacterial quantification should be apparent across time point groups compared to the zero hour for smooth and rough. It may be better to provide statistical analysis and P values for the non-normalised data, maybe removing 1B. This may result in loss of statistical significance

as there doesn't seem to be a hugely convincing difference between the smooth and rough, until the 28 day time point.

- Also, you may want to reconsider your choice of statistical test, perhaps a T-test is more appropriate, although I am not an expert statistician.
- Finally I would recommend changing the colours from red and green to make these figures accessible to red-green colour blind individuals.

Figure 1C:

- It could be useful to mention the use of *M. abscessus* fluorescent strains in the results paragraph, and also would be of interest to note if there were any problems with plasmid maintenance in these *M. abscessus* strains during the time course?
- It is unclear what figure 1C is trying to convey, as the empty arrowheads indicate loose *M. abscessus* lesions, however there is no visible tomato fluorescence in 1Ci.

Figure 1D is not referenced in the manuscript. I think it would be useful to slightly expand the results section "Adult zebrafish mount a robust inflammatory response to *M. abscessus* infection" to describe figures 1C and 1D, what they are showing and what conclusions are being drawn from them.

Figure 2 - what anatomic regions were granulomas imaged in and being found in?

Figure 3 is very interesting. It is possibly worth suggesting that the increase in smooth organised granulomas at 28 dpi may be due to a phenotypic switch to the rough morphotype? It is noted in the discussion that no phenotype switching was observed. Including a note about this possibility and that it was not seen may be beneficial in this results section.

Figure 4 - consider including supplemental figure 2 (mainly 2B) in the main figure, as this is very interesting showing the ratio of T cell GFP to *M. abscessus* tomato fluorescence for the smooth and rough morphotypes.

Figure 5:

- Overall, I really like figure 5, it shows a convincing role of T cells in control of *M. abscessus*, especially for control of the rough morphotype over the smooth. Would be useful to know if WT zebrafish do eventually die of *M. abscessus* infection earlier than uninfected animals. This experiment may be too long to complete to include in this paper and isn't necessary, but including a note in the manuscript if it is known when *M. abscessus* infected WT adult zebrafish die would be very interesting.
- Again it may make the figure more accessible by changing the colours from red and green.

Figure 6A - schematic would be clearer by showing injection of wasabi or tomato abscessus into zebrafish, then onto plates, and it needs to be properly described in the figure legend. Currently the schematic is not adding any value to the figure or manuscript.

Reviewer #1 (Remarks to the Author):

I consider this as an interesting paper, which utilizes zebrafish model to study pathophysiology of smooth (S) and rough (R) variants of *M. abscessus*. This information is relevant as at the current stage, the knowledge of interaction between these different *M. abscessus* variants and immune cells *in vivo* is limited.

Earlier, zebrafish model has been used successfully to study other mycobacterial infection. The model is well chosen as it allows using various mutant lines and advanced microscopy. In addition, ethically this model is sound. In this current manuscript, methods are solid and well described. The paper is logically written and thus easy to follow. In conclusion, obtained results provide evidence for usage of zebrafish model for studying the *M. abscessus* infection.

General comments:

Although I do not have major criticism about the experiments, I think that technically the paper is somewhat limited. To make the paper stronger it would have been interesting to know if mechanisms regulating switching between R and S could be studied in this fish model.

We have concentrated our efforts on investigating host determinants of infection outcome as this is our strength. We have not observed S to R switching in our experiments and refer Reviewer 1 to our response to Reviewer 4 comment about the possibilities of using our model to investigate transcriptional phenotype switching.

Reviewer #2 (Remarks to the Author):

The authors report on an adult zebrafish model for *Mycobacterium abscessus*, a notoriously difficult to treat pathogen associated with high mortality in specific groups, such as those with cystic fibrosis. In human infections, a stark difference has been observed between two morphotypes, rough and smooth based on the presence or absence of a surface glycopeptidolipid. The smooth morphotype possesses it and are more environmental and less invasive. The more virulent rough strain emerges during the course of infection in humans and is associated with more severe disease. The rough strain also survives in human monocytes in culture versus the smooth strain which is cleared, and the same is true for a lung infection model.

While the subject matter is interesting and important, this paper does not shed light on it. Unfortunately it presents a mishmash of poorly done, incorrectly analyzed experiments with no real interpretation.

We hope that our new experiments and manuscript reorganisation satisfy the reviewer's wide-ranging criticisms.

The first set of findings contradict the literature. The smooth strain appears to grow better in the zebrafish than the rough strain and forms more necrotic foci (Figure 1D versus 1C, even if the authors do not interpret it that way). This difference from the literature is not commented upon and rather the example granulomas presented in Figure 2 appear to contradict this results as the one with the most bacteria is the rough one (Figure 2A top, left panel).

We agree that our findings are contradictory to the mouse literature and reiterate that that is the major advance of our model. In comparison to immunocompetent mice, which naturally clear the infection in 4 weeks (Bernut et al. 2014, *Antimicrob Agents Chemother* 58:4054-63), our platform maintains bacterial burden for an extended period of time (>40 days), allowing to successfully establish chronic infections with physiopathological granulomas, which appear very similar to the ones observed in humans infected with *M. abscessus*. We also argue that our results for the R strain histopathology are consistent with the human literature as it drives more granuloma necrosis than the S strain. We do not find any *in vivo* human literature that contradicts the prolonged survival of S strain our model, in fact extended survival

of the S strain is a very likely prerequisite for the mutational conversion of S bacteria to the R phenotype, affecting usually the biosynthesis and/or transport machinery of GPL.

Figures 1C and 1D (now replaced by Figures 3A and B) were picked as illustrations of *tnfa* promoter induction, not for to be representative of the heterogenous range of granuloma morphologies across 4 conditions (2 strains and 2 timepoints). We routinely identified 10+ lesions per animal so it is impossible to estimate bacterial burden or rate of granuloma necrosis from a single image of a single granuloma from a single animal as the reviewer is suggesting from their interpretation of these panels. As part of our manuscript reorganisation these panels now fall after the granuloma morphology census data in Figure 2.

The oil red stains in panels B and C don't provide much information on the differences.

Added additional text "illustrating a potential difference in granuloma macrophage physiology between R and *S. M. abscessus*" to our results for what is now Fig 2B and C.

The authors try to categorize lesions as organized and not organized in Figure 3 but this does not make much sense from the histology shown, and the comparisons shown of the quantitations do not either.

[This quantification is now located in Fig 2D and 2E, and the same quantification methodology is used in the new Figure 3]

We categorised lesions based on the presence of necrotic cores and directional host nucleus flattening. The quantifications are a census of all sites with bacterial from serial cryosections across entire animals. We present these census data two ways: 1) a per lesion comparison where 25% means 25% of all granulomas in an animal were organised, and 2) a bacterial burden distribution comparison where 25% means 25% of all the imaged bacteria in an animal were within organised granulomas. We had hoped that this was apparent from our figures, legends, associated body text, and references but are happy to add additional phrasing if the reviewer has specific suggestions to clarify these points.

In Figure 5, the authors find that animals succumb to both strains if they lack T cells, which is not surprising. However, they claim that "T cells are necessary to control the R strain but not the strain", a claim that is unsubstantiated by their findings. Their survival curve (panel A) in the T cell deficient (*lck*^{-/-}) animals are no different for the S and R, despite a barely significant p value provided by the authors. Their interpretation of panel B is also incorrect. Despite their finding a p value difference for the R and not the S strain, between wildtype and *lck*^{-/-} fish, the spread is large and the sample size small (3 fish). Also panel B is displayed in a strange way. One simply cannot conclude anything from this experiment. Even if true, what does the finding mean for pathogenesis.

[This data set is now located in Figure 4D and 4E]

The CFU experiment dataset is of three independent replicates each with multiple animals per condition as described in the figure legend. Taken together with our new dexamethasone and *tnfa* knockdown experiments in Figure 3, we now provide a much more compelling differentiation between the R strain – requires host TNF to drive the formation of necrotic granulomas and is controlled by T cells at the 2 week timepoint, and the S strain – targeted by host TNF but does not interact with T cells at the 2 week timepoint. Our mixed infection experiments further support these assertions as we show the R strain activates dexamethasone-sensitive inflammation that targets the S strain while T cell deficiency does not affect the mixed infection-induced restraint of the S strain.

Similarly, in the next set of results, where they examine mixed infections, the authors claim that the increased growth of R is attenuated by co-infection with S. But this is from two animals and the spread in the S only is large. This needs to be repeated and if true, an interpretation of the results is needed.

This dataset is from two independent replicates each with multiple animals as described in the figure legend. Each experiment produced a statistically significant difference between S recovered from single and mixed infection animals.

In summary, this paper suffers from being technically flawed, with flawed statistical analyses, and no interpretation of the data with regard to how and why they differ from the literature and are internally contradictory as well.

The existing *M. abscessus* pathogenesis literature has been largely written using immunocompromised mice and zebrafish embryos which lack an adaptive immune system. It is unsurprising that our study of persistent infections in fully immunocompetent zebrafish, which are themselves natural hosts for other NTMs, will challenge or differ from the existing literature.

We have made a concerted effort to clarify our technical and statistical approaches, and more clearly integrate our findings into what is known from human disease. Our major mechanistic findings that 1) TNF drives granuloma expansion by the hyperinflammatory rough variant while controlling smooth variant, and 2) T cell mediated immunity is more active against the rough than the smooth variant, represent important advances in investigating the pathobiology of persistent *M. abscessus* infection in animal models. These particular investigations cannot be easily carried out in existing mouse models of *M. abscessus* infection.

The mixed R and S infections are a novel approach to examine what happens in humans where both isolates have been identified in the same patient as either coming from co-infections with different clones or after S-to-R transition events. Our finding that invasive R infection antagonises T cell-independent immunity into controlling S bacteria both confirms our major mechanistic findings and provides platform for further study of this interesting tripartite set of clinically-relevant host-pathogen interactions.

Reviewer #3 (Remarks to the Author):

In this submission, the authors describe using a zebrafish model to examine differences in host immune response resulting from infection with different Mycobacterium abscessus variants. Their approach is thorough, evaluating both single and mixed infections, with appropriate endpoints of histology/fluorescence and bacterial culture. I found the paper to be generally well written. At times the results section strayed into discussion topics, but this fit the flow and may be acceptable for this journal. I do have a few minor comments as follows:

L227 - If S variant dropped, why is the ratio suggesting a greater proportion relative to R (0.5:1, R:S), when R had similar recoveries in single or mixed?

We have since removed this ratio calculation as it was confusing to other readers. Briefly, although the S variant burden was dropped in mixed infections (apparent on a Log scale), it was still maintained at a higher burden than the R from any condition (apparent on a linear scale).

L265 - would the T cell depletion associated with AIDS not benefit *M. abscessus*? You saw greater burdens of R variant in T cell deficient zfish.

The lack of epidemiological association between HIV/AIDS and *M. abscessus* infection susceptibility is indeed puzzling. It seems unlikely that patients are not exposed to *M. abscessus* as there is geographical overlap, especially across South East Asia [PMID: 30689814, 22345581]. The most likely explanation that we are aware of is that a significant proportion of *M. abscessus* (and indeed other NTM) infections are misdiagnosed as *Mycobacterium tuberculosis*, fail to respond to anti-tuberculosis antibiotics and are never correctly classified.

We address the implications of our findings in the paragraph 4/line 308 of the discussion arguing that if HIV patients encounter S variant *M. abscessus*, they would be able to control infection just as well as HIV-negative individuals as immunological control of the S variant was T cell-independent in our model.

L340 - more detail is required for basic husbandry conditions. Such information is critical for reproducibility. What were the fish fed and how often? What are the water parameters (conductivity, pH)? Static tanks? Recirc? Flow through? Stocking density?

How many animals held in each beaker following injection?

L359 - What dose of tricaine?

L360 - how was the homogenization done? Chemical or physical? Explain.

We have added additional information to methods section and references to our step by step methods paper detailing these procedures.

Reviewer #4 (Remarks to the Author):

This paper introduces the potential of adult zebrafish as a model of chronic *M. abscessus* infection. This is very interesting due to the lack of suitable models for studying the adaptive immune response to *M. abscessus*. The authors show differences in granuloma formation between smooth and rough *M. abscessus* morphotypes. They also show the importance of T cells in controlling rough *M. abscessus* infection, whereas the smooth morphotype appears to not induce the same T cell response. This very nicely brings together previous contradicting work regarding innate immune responses to smooth and rough morphotypes and clinical observations. Overall, I found the paper to be very interesting, showing novel work using adult zebrafish, and generally convincing. It is well written and clear to follow, with good referencing to the wider literature. I would recommend it be accepted for publication following some amendments (most importantly figures 1A and 1B).

Introduction - I would suggest making note of the idea that while an irreversible genetic switch can occur from smooth to rough, GPL production can also be controlled on a variable transcriptional scale between the smooth and rough morphotypes.

We have modified text in the introduction to “It is well established that these morphological differences between S and R variants are dependent on the presence or absence of surface-exposed glycopeptidolipids (GPL), respectively, which are reduced or lost in R variants by transcription downregulation or loss of function mutations” and added references. Also, please see our response below to the related Figure 3 comments.

Figures 1A and 1B: I have the following comments

- I wonder if these would be better presented as box plots pooling all animals from across experimental repeats, rather than line graphs.

We believe that a strength of the line graph presentation in Figure 1A is to illustrate that, across a range of input infection doses, infections with R or S follow distinct trajectories with R burdens maintaining while S burdens increase.

- I don't see the value in normalising the quantification data to the zero hour time points in Figure 1B. Changes in bacterial quantification should be apparent across time point groups compared to the zero hour for smooth and rough. It may be better to provide statistical analysis and P values for the non-normalised data, maybe removing 1B. This may result in loss of statistical significance as there doesn't seem to be a hugely convincing difference between the smooth and rough, until the 28 day time point.

Following on from our previous response, the normalised presentation in Figure 1B is to illustrate the general trends of persistent R burdens and increasing S burdens independent of input infection dose.

- Also, you may want to reconsider your choice of statistical test, perhaps a T-test is more appropriate, although I am not an expert statistician.

We had initially used a T test as we thought we were only making pairwise comparisons at single timepoints but we told to use this multiple comparison test by a previous reviewer. As would be expected, more of the R vs S comparisons were statistically significant when analysed by T test. We are comfortable with the more rigorous ANOVA test to err on the conservative side.

- Finally I would recommend changing the colours from red and green to make these figures accessible to red-green colour blind individuals.

We have altered our pseudocolouring to change any red to magenta in our images and graphs.

Figure 1C:

- It could be useful to mention the use of *M. abscessus* fluorescent strains in the results paragraph, and also would be of interest to note if there were any problems with plasmid maintenance in these *M. abscessus* strains during the time course?

We have added mention of the use of plasmid-based fluorescent proteins in the results paragraph (line 128). We did not perform any direct analysis of plasmid maintenance but plasmid loss is a known phenomenon with *M. abscessus*. As with our experience from the use of these same plasmids with long term *M. marinum* infection experiments, it is exceedingly rare to observe granulomas without mycobacterial fluorescence at the timepoints assayed including our longest 70 dpi timepoint (Supplementary Figure 1).

- It is unclear what figure 1C is trying to convey, as the empty arrowheads indicate loose *M. abscessus* lesions, however there is no visible tomato fluorescence in 1Ci.

We replaced these images with different images in Figure 3 showing very specific *tnaA::gfp* expression despite containing relatively few *M. abscessus* R bacteria. We hope that our colouring of *M. abscessus* to magenta makes it easier to see the bacterial fluorescence in these images.

Figure 1D is not referenced in the manuscript. I think it would be useful to slightly expand the results section "Adult zebrafish mount a robust inflammatory response to *M. abscessus* infection" to describe figures 1C and 1D, what they are showing and what conclusions are being drawn from them.

We have expanded the results section to investigate TNF signalling in a new Figure 3 using dexamethasone and Crispr-Cas9 knockdown of host TNF to show a novel pathogenic role for TNF in R pathogenesis which contrasts with the expected protective role for TNF against S infection.

Figure 2 - what anatomic regions were granulomas imaged in and being found in?

Granulomas were only found within the abdominal cavity following intraperitoneal injections. We have added this note in the text (line 131).

Figure 3 is very interesting. It is possibly worth suggesting that the increase in smooth organised granulomas at 28 dpi may be due to a phenotypic switch to the rough morphotype? It is noted in the discussion that no phenotype switching was observed. Including a note about this possibility and that it was not seen may be beneficial in this results section.

The transcriptional downregulation of cell surface elements during infection stress is certainly a possibility for explaining the "caught up" virulence of the smooth strain by 28 dpi. However, we were unable to accurately quantify bacterial gene expression from our homogenised samples to examine this possibility. Thus we have added a note on this possibility in the discussion (line 333).

We have data showing the smooth strain can undergo reversible phenotypic transition to a rough-like morphology when recovered from adult zebrafish infections onto hygromycin-containing agar that is not present when passaging from frozen stocks or liquid media outgrowths. This phenotype reverses back to a smooth morphology upon passage onto media with either no antibiotics or the same high concentration of hygromycin. External factors, such as sub-inhibitory antibiotic concentrations, are known to promote a transient S-to-R phenotypic change with more aggregated cultures, resistance to phagocytosis, and decreased GPL resulting from down-regulation of GPL biosynthetic genes (Tsai et al doi: 10.1128/AAC.01132-15 and Lee et al. doi: 10.1007/s12275-017-6503-7).

Figure 4 - consider including supplemental figure 2 (mainly 2B) in the main figure, as this is very interesting showing the ratio of T cell GFP to *M. abscessus* tomato fluorescence for the smooth and rough morphotypes.

Supplemental Figure 2B has been incorporated into the main figure as Figure 4C as suggested.

Figure 5:

- Overall, I really like figure 5, it shows a convincing role of T cells in control of *M. abscessus*, especially for control of the rough morphotype over the smooth. Would be useful to know if WT zebrafish do eventually die of *M. abscessus* infection earlier than uninfected animals. This experiment may be too long to complete to include in this paper and isn't necessary, but including a note in the manuscript if it is known when *M. abscessus* infected WT adult zebrafish die would be very interesting.

WT zebrafish seem quite “happy” carrying *M. abscessus* infection and seem to deal as well as or even better than chronic *M. marinum* infections in our hands. We have made mention of this with reference to Supplementary Figure 1 with the 70 dpi WT zebrafish having clear (and fluorescent) granulomas despite appearing outwardly healthy (line 139).

- Again it may make the figure more accessible by changing the colours from red and green.

Red has been recoloured as magenta

Figure 6A - schematic would be clearer by showing injection of wasabi or tomato abscessus into zebrafish, then onto plates, and it needs to be properly described in the figure legend. Currently the schematic is not adding any value to the figure or manuscript.

Thank you for pointing this out. In hindsight the schematic makes much more sense with the positions of the fish and bacteria reversed!

REVIEWER COMMENTS

Reviewer #1 (Remarks to the Author):

Although it appears that there is no switch between R and S phenotypes in this model I still consider this model important and thus I would favour acceptance.

[After being asked to provide feedback on the comments made by reviewer #2]

As commented earlier, I do feel that this paper is acceptable in its current form.

Related to reviewer 2 comments, I think that the quality of the Figure 2 is appropriate and thus I trust the data. When using adult zebrafish as a model one tries to minimize the number of animals used and thus I would be hesitant to ask additional experiments. What comes to surprising results, that happens sometimes and it does not make the results less important.

Reviewer #2 (Remarks to the Author):

I appreciate the authors' attempts to revise the paper. I tried to take a fresh look at the paper and fear that the revised paper does not offer the insights that the authors would hope for. The conclusions are often not justified, and in many cases, may be based on noise from small numbers. Unexpected results are not explained and are unconvincing. Granuloma microscopic classifications are unconvincing. As a result, there is little light shed on R and S strain pathogenesis or differences, nor on other aspects of *M. abscessus* pathogenesis. Unfortunately, the claim in the last sentence of the abstract "In this work, we ...provide insight into the immunopathogenesis of chronic *M. abscessus* infection" is simply not supported by the data.

Detailed comments:

Despite small numbers, it does appear that the S strain achieves higher burdens than the R strain. However, the authors then suggest unconvincingly in Figure 2A that the R strain forms more organized granulomas than the S strain and that the R bacteria lie within granulomas whereas the S bacteria lie largely free with only a few bacteria in the granulomas. These data are not convincing. First, the middle panel where they call a 14-day S granuloma unorganized looks organized, just not necrotic. As for the 28-day S granuloma that they call organized – it is not clear that it has any fewer bacteria than the very large R granuloma just above it. The staining just looks dimmer. The quantitation that follows is suspect because the examples are not at all convincing.

In Figure 3, the authors find that dexamethasone increases bacterial counts in both S and R. This is expected. But they then report that TNF deficiency increases S burdens but decreases R burdens. Why? Particularly since dexamethasone decreases TNF, presumably in both R and S infections (though the authors do not specify which strain they used in Figure 3c). This is a very surprising result, and the authors provide no convincing explanation. Yet, T cell deficiency has the same effect on bacterial burdens in both strains.

Figure 4D, the only survival experiment, is telling because there is no mortality with either the R or S strain with inocula used – how many bacteria, I could not find in the main text, legend, or methods. Do the authors find this surprising? Can they find an inoculum which separates the R and S responses?

Have the authors considered testing the TNF mutants with the mixed experiments to see if the opposite effects on R and S hold up?

The authors may want to consider stepping back and trying to look at the data again, increasing the number of animals used, and getting independent evaluation of their histology to try to understand the model better. At the present time, unfortunately, it is not clear that this work adds to the already confusing field. Rather it sows more confusion.

Reviewer #3 (Remarks to the Author):

The authors addressed this reviewer's previous concerns.

Reviewer #4 (Remarks to the Author):

All my points to the authors have been adequately addressed.

I found this revised manuscript to be improved, providing interesting and important analysis on the use of adult zebrafish as a model for chronic *M. abscessus* infection and the differing immune responses in controlling the smooth and rough morphotypes. I think it will be of interest to others in the field and lead to future developments, potentially using this model to study morphotype switching in *M. abscessus*, and in further study of immune responses to these morphotypes.

I found the roles of both TNF and T cells in controlling rough pathogenesis more than smooth to be consistent with the concept that the smooth morphotype is less immunostimulatory, leading to a more persistent infection as opposed to the more pathogenic rough morphotype, as is observed clinically and in other animal models. This is further supported by the mixed infection model, with the immune response generated by the presence of the rough morphotype subsequently also restricting growth of the smooth. While questions remain about the possible role of in vivo morphotype switching, this has been discussed by the authors and is a potential future application of this model. Studying in vivo morphotype switching will be challenging, due to difficulties in identifying morphotypes in vivo and subsequent switching or reversion in vitro.

Minor comments:

I would suggest possibly adding a few sentences to the methods section briefly explaining how *M. abscessus* CFU was determined for the initial inoculum, however this is not essential as it seems to be quite an intensive method, and I did not mention this in the initial review.

We thank reviewers 1,3, and 4 for their positive comments. In response to reviewer 4 I have added the method used to determine initial inoculum. We respond to Reviewer 2's comments in full below:

Line numbers refer to the tracked changes version of the manuscript

Reviewer #2 (Remarks to the Author):

I appreciate the authors' attempts to revise the paper. I tried to take a fresh look at the paper and fear that the revised paper does not offer the insights that the authors would hope for. The conclusions are often not justified, and in many cases, may be based on noise from small numbers.

Without specific direction it is hard to respond to this criticism. I believe we have previously pointed out Figure 5E plots averages of experiments, each with multiple animals.

Unexpected results are not explained and are unconvincing. Granuloma microscopic classifications are unconvincing.

I have counted new datasets and classified granulomas as necrotic or cellular (non-necrotic). This is a much clearer definition that we have used in the zebrafish-*Mycobacterium model* (eg Oehlers *Nature* 2015, Cronan *Immunity* 2016, Cronan *Cell* 2021) but does not affect the previous classifications of granulomas as necrosis was the key determining factor. As shown below, the results of the new dataset entirely recapitulate the key points of the old figure 2 that there is delayed granuloma necrosis in S infection at 2 wpi and that this difference is not statistically distinguishable at 4 wpi. I have replaced the graphs in Figures 1 D and E with the combined dataset, and updated text in the section beginning Line 142.

New dataset

Old dataset

Combined datasets

If the "unexpected results" refers to the *tnfa* KD being protective against *M. abscessus* R infection, we have performed additional replicates of the *tnfa* knockdown experiment, and provide new data with *tnfr1* and *tnfr2* knockdown confirming our unexpected *tnfa* KD/*M. abscessus* R phenotype (Figure 4). We further tie in our T cell phenotypes by showing that the Treg subset limit *M. abscessus* R growth (Figure 6). Overall we provide 3 lines of evidence that *M. abscessus* R persistence is supported by host immunopathogenesis, and 2 lines of evidence connecting *M.*

abscessus R pathogenesis to the formation of necrotic granulomas that act as a niche for bacterial growth.

As a result, there is little light shed on R and S strain pathogenesis or differences, nor on other aspects of *M. abscessus* pathogenesis. Unfortunately, the claim in the last sentence of the abstract “In this work, we ...provide insight into the immunopathogenesis of chronic *M. abscessus* infection” is simply not supported by the data.

I have removed “and provide insight into the immunopathogenesis of chronic *M. abscessus* infection” from the abstract as requested.

Detailed comments:

Despite small numbers, it does appear that the S strain achieves higher burdens than the R strain. However, the authors then suggest unconvincingly in Figure 2A that the R strain forms more organized granulomas than the S strain and that the R bacteria lie within granulomas whereas the S bacteria lie largely free with only a few bacteria in the granulomas. These data are not convincing. First, the middle panel where they call a 14-day S granuloma unorganized looks organized, just not necrotic. As for the 28-day S granuloma that they call organized – it is not clear that it has any fewer bacteria than the very large R granuloma just above it. The staining just looks dimmer.

The quantification shown in 2E represent totals censuses of individual fish. Each fish had multiple granulomas (ranging from 10s to over 100 in some 28 dpi animals). The example granulomas selected in Figure 2A are meant to be examples of granuloma morphology not individual examples of the numerator used in the calculations to generate Figure 2E.

The quantitation that follows is suspect because the examples are not at all convincing.

I have counted new datasets (above) and recounted the existing datasets to classify granulomas as necrotic or cellular (non-necrotic).

In Figure 3, the authors find that dexamethasone increases bacterial counts in both S and R. This is expected. But they then report that TNF deficiency increases S burdens but decreases R burdens. Why? Particularly since dexamethasone decreases TNF, presumably in both R and S infections (though the authors do not specify which strain they used in Figure 3c). This is a very surprising result, and the authors provide no convincing explanation.

Dexamethasone has many immunosuppressive effects beyond TNF so this effect is hardly surprising eg Grab et al *Nature Communications* 2019: TNF/TNFR-independent suppression of mycobacterial infection-induced cell death by corticosteroids. This effect has recently been acutely illustrated by the differential susceptibility of IBD patients on corticosteroids vs TNF antagonists to COVID-19 (Brenner et al *Gastroenterology* 2020).

I have reworded the discussion starting line 342 to emphasise this point.

Yet, T cell deficiency has the same effect on bacterial burdens in both strains.

Figure 5D and E (around line 250) explain that there is a stronger effect of T cell deficiency on the immune control of R strain compared to S strain.

Figure 4D, the only survival experiment, is telling because there is no mortality with either the R or S strain with inocula used – how many bacteria, I could not find in the main text, legend, or methods. Do the authors find this surprising? Can they find an inoculum which separates the R and S responses?

This experiment was carried out with our standard dose of 10^5 CFU/animal as detailed in Line 354-356 of the methods.

We would require >6 months and external research funding to resurrect the lck mutant colony to repeat this experiment. There are also ethical considerations to reproducing this survival experiment now that we know there is increased mortality in T cell-deficient animals.

The alternative experiment of increasing infectious doses in WT animals has already been carried out with both strains. Up to 10^6 CFU/fish caused negligible mortality. We settled on (and report) 10^5 CFU for our routine experiments as this scaled more appropriately from the 10^6 - 10^7 CFU used to infect mice.

Have the authors considered testing the TNF mutants with the mixed experiments to see if the opposite effects on R and S hold up?

We have performed additional replicates of the mixed experiment with both wildtype animals and *tnfa* crispr-cas9 injection on the mixed infection phenotype. This produced very difficult datasets to assess with unexpected results from the scrambled control animals. In the interests of clarity, we have removed all mixed infection data and focussed on the host immunopathology elements of our manuscript.

The authors may want to consider stepping back and trying to look at the data again, increasing the number of animals used, and getting independent evaluation of their histology to try to understand the model better. At the present time, unfortunately, it is not clear that this work adds to the already confusing field. Rather it sows more confusion.

In the interests of clarity we have removed all mixed infection data and focussed on the host immunopathology elements of our manuscript. We have performed additional replicates of the *tnfa* knockdown experiment, and provide new data with *tnfr1* and *tnfr2* knockdown confirming our unexpected *tnfa* KD/*M. abscessus* R phenotype (Figure 4). We further tie in our T cell phenotypes by showing that the Treg subset limit *M. abscessus* R growth (Figure 6). Overall we provide 3 lines of evidence that *M. abscessus* R persistence is supported by host immunopathogenesis, and 2 lines of evidence connecting *M. abscessus* R pathogenesis to the formation of necrotic granulomas that act as a niche for bacterial growth.

REVIEWERS' COMMENTS

Reviewer #2 (Remarks to the Author):

The authors have responded satisfactorily and the paper will make a useful addition to the field. I have one remaining request. They should not use P values where multiple comparisons are made. Rather an ANOVA analysis with a post hoc analysis should be done and the results reported as the standardized asterisks. Non significance should be denoted NS or something like that. This should be done for the figure shown in the response letter and in any other relevant figures in the paper.

Response to reviewer

Reviewer #2 (Remarks to the Author):

The authors have responded satisfactorily and the paper will make a useful addition to the field. I have one remaining request. They should not use P values where multiple comparisons are made. Rather an ANOVA analysis with a post hoc analysis should be done and the results reported as the standardized asterisks. Non significance should be denoted NS or something like that. This should be done for the figure shown in the response letter and in any other relevant figures in the paper.

We had already performed ANOVA with post hoc analyses for datasets with three or more groups and are able to provide precise P values from these tests using commercially available Graphpad Prism. The reviewer's requested use of asterisks to replace precise P values conflicts with editorial directions from the journal office so we have continued to use the precise P values provided by our ANOVA with post hoc analyses.